# Phenolic Fingerprinting, Antioxidant, and Deterrent Potentials of *Persicaria maculosa* Extracts

**DOI:** 10.3390/molecules25133054

**Published:** 2020-07-03

**Authors:** Luisa Quesada-Romero, Carlos Fernández-Galleguillos, Jan Bergmann, María-Eugenia Amorós, Felipe Jiménez-Aspee, Andrés González, Mario Simirgiotis, Carmen Rossini

**Affiliations:** 1Laboratorio de Ecología Química, Instituto de Química, Pontificia Universidad Católica de Valparaíso, Avda. Universidad 330. Curauma, Valparaíso 2340000, Chile; lufequro@gmail.com (L.Q.-R.); jan.bergmann@pucv.cl (J.B.); 2Facultad de Ciencias para el cuidado de la Salud, Universidad San Sebastián, General Lagos 1163, Valdivia 5090000, Chile; 3Instituto de Farmacia, Facultad de Ciencias, Universidad Austral de Chile, Valdivia 5090000, Chile; carlos.fernandez@uach.cl; 4Laboratorio de Ecología Química, Facultad de Química, Universidad de la República, Gral. Flores 2124 CP 11800, Montevideo 11800, Uruguay; maruamoros@gmail.com (M.-E.A.); agonzal@fq.edu.uy (A.G.); 5Departamento de Ciencias Básicas Biomédicas, Facultad de Ciencias de la Salud, Universidad de Talca, Avenida Lircay S/N, Talca 3460000, Chile; fjimenez@utalca.cl; 6Center for Interdisciplinary Studies on the Nervous System, Universidad Austral de Chile, Campus Isla Teja, Valdivia 5090000, Chile

**Keywords:** antioxidant, flavonoids, polygonaceae, HPLC-ESI-MS/MS, antifeedant activity

## Abstract

*Persicaria maculosa* (Polygonaceae) (known as lady’s thumb) is an annual morphologically variable weed that is widely distributed in Chile. The purpose of this study was to investigate the antifeedant potential of methanolic (MeOH), ethanolic (EtOH), and dichloromethane (DCM) extracts from the aerial parts of this plant collected in the Valparaíso and Curicó provinces (Chile) and relate this activity to the antioxidant capacity and the presence of phenolic compounds in the extracts. A phenolic profile based on HPLC-ESI-MS/MS allowed the identification of 26 phenolic compounds, most of them glycosyl derivatives of isorhamnetin, quercetin, and kaempferol. In addition, the total phenolic content (TP), total flavonoids (TF), and antioxidant activity measured by 2,2-diphenyl-1-picrylhydrazyl (DPPH), superoxide anion scavenging (O_2_^−^), ferric-reducing antioxidant power (FRAP), and cupric-reducing antioxidant capacity (CUPRAC) of the extracts are reported. The antifeedant potentials of the plant extracts were tested against *Epilachna paenulata*, *Pseudaletia adultera*, *Macrosiphum euphorbiae*, and *Diaphorina citri* insects for the first time. The activity against the aphid *M. euphorbiae* was significant for the DCM extracts of plants from Valparaíso and Curicó (settling % = 23% ± 4% and 23% ± 5%, respectively). The antifeedant activities against the beetle *E. paenulata* and the lepidoptera *P. adultera* were significant for Valparaíso extracts, especially when tested against *E. Paenulata* (IFP = 1.0 ± 0.0). Finally, the MeOH and EtOH extracts from Valparaíso plants reduced the diet consumption of the psilid *D. citri* (*p* < 0.05). The results showed that *P. maculosa* is a good source of flavonoids with some antioxidant capacities and has potential interest as botanical eco-friendly alternative with deterrent activity.

## 1. Introduction

The Polygonaceae family is represented by approximately 1300 species and 59 genera distributed around the world in temperate regions [1]. Particularly in Chile, the Polygonaceae family is represented by seven genera and 40 perennial species that are invasive in most geographical regions. Members of this family have been used in traditional medicine to treat several diseases [2]. Different chemical constituents in these plants have been reported, such as triterpenoids [3], sesquiterpenoids [4], coumarins [5], anthraquinones [6], phenylpropanoids [7], tannins [8], lignans [9], and flavonoids [10,11]. Amongst them, flavonoids are groups of the most common compounds found in the Polygonaceae species, playing an important role as chemotaxonomic markers [12].

Flavonoids are interesting secondary metabolites due to their benefits for human health, particularly for the prevention of the oxidative stress process related to human diseases [13]. Flavonoids are metabolites used frequently by cells for the protection against the harmful effects of reactive oxygen (ROS) species. In addition, these antioxidant compounds can control agricultural pests due to their effective protective role against insects [14].

Many studies have been reported on the insecticidal properties of the *Persicaria* species. *Persicaria hydropiper* (Synonym: *Polygonum hydropiper*) is the most studied species due to its interesting phenolic constituents and interesting biological potential [15]; this plant showed deterrent activity against the third instar larvae of *Spilarctia obliqua* and *Spodoptera litura* [16]. In addition, the acaricidal activity was observed by the petroleum ether and acetone extracts of this plant against the tea red spider mite (*Oligonychus coffeae*) [17]. On the other hand, *Persicaria maculosa* (formerly *Polygonum persicaria*) is an annual morphologically variable weed known as lady’s thumb that is widely distributed. Insecticidal activity of crude extracts of *P. maculosa* against the red flour beetle (*Tribolium castaneum*) was reported [18], and recently lectin extracts from this plant have been found to affect the antioxidant system of the cotton bollworm (*Helicoverpa armigera*) [19]. Previous studies report several flavonoids in the aerial parts of *P. maculosa* [20,21,22].

In this study, we report the chemical profiling of two populations of *P. maculosa* extracts by High-performance liquid chromatography/electrospray ionization tandem mass spectrometry (HPLC-ESI-MS/MS) a powerful tool used to tentatively identify structures of flavonoids and other phenolics compounds in extracts from natural plant sources based on deprotonated parent ions and characteristic daughter fragments [23]. Antioxidant capacities and phenolic and flavonoid content were also determined. Additionally, four insect species (*Epilachna paenulata*, *Pseudaletia adultera*, *Macrosiphum euphorbiae*, and *Diaphorina citri*) were chosen to study the potential deterrent activities based on the different feeding habits, diet breadthand host plants.

## 2. Results and Discussion

### 2.1. Total Phenolic (TP) and Total Flavonoid (TF) Content and Antioxidant Activity

In the present study, *P. maculosa* polar extracts (EtOH and MeOH) and non-polar extract (DCM) were studied for their contents of total phenolics (TP) and total flavonoids (TF) and their antioxidant activity. The dichloromethane (DCM) extract did not present a TP or TF content and was inactive in all the antioxidant assays carried out. The extraction yields for the Valparaíso extracts were 7.1% for MeOH, 1.9% for EtOH and 0.3% for DCM; and for the Curicó extracts, 7.4% for MeOH, 2.3% for EtOH and 0.3% for DCM. Our results agree with other studies that reported the effect of solvent polarity on the extraction yields and antioxidant activity [24,25].

The TP content was significantly higher in the EtOH extract from Curicó, while the lowest content was found in the MeOH extract from Valparaíso (Table 1). No significant differences were observed between the EtOH and MeOH extracts from Valparaíso and Curicó in TP content. The lowest TF content was found in the EtOH extract from Valparaíso, while the other samples showed no significant differences (Table 1, one-way ANOVA followed by Tukey’s test (significant differences are reported when *p* < 0.05).

For the antioxidant activity, four different chemical-based methodologies were carried out, including the scavenging of free radicals 2,2-diphenyl-1-picrylhydrazyl (DPPH) and superoxide anions (O_2_^−^), and the reducing capacity of the extracts ferric-reducing antioxidant power (FRAP) and cupric ion-reducing antioxidant capacity (CUPRAC) assays. Since the combination of several antioxidant assays allows a more complete evaluation of the antioxidant properties of a sample, these assays provide complementary information about the interaction between radicals and samples [26]. In the free radical scavenging assays, the results were expressed as the concentration of extract that scavenged the free radical by 50% (SC_50_), and the highest antioxidant activity was found in the EtOH extract collected in Curicó (SC_50_ 12.5 μg/mL), followed by the EtOH extract from Valparaíso (Table 2). The MeOH extracts collected in Valparaíso showed the best reducing power in the CUPRAC assay (1.1 mmol TE/g extract), followed by the EtOH extract from Curicó (0.8 mmol TE/g extract). These two samples were also the most active in the FRAP assay (Table 2). In the literature, the antioxidant activity of the MeOH extracts of the roots, stems, leaves, and flowers of *Polygonum sachalinensis* was carried out by the DPPH assay, yielding SC_50_ values between 20 and 85 μg/mL, with the highest antioxidant effect for the flower and leaf extracts and the lowest for the root and stem extracts [27]. In another report, several fractions of *Persicaria hydropiper* showed an antioxidant activity (DPPH assay) in the range of IC_50_ 13.3–93.8 μg/mL, with the strongest activity in the ethyl acetate extract [28]. The flavonoid content and composition of the leaves of *Polygonum hydropiper* L. was correlated with the antioxidant activity found in the Trolox equivalent antioxidant capacity (TEAC) assay [29]. On the other hand, the Pearson’s correlation coefficient in our samples did not show a significant correlation between the TF content and any of the antioxidant assays. A strong but not significant correlation (*r* = −0.9451, *p* > 0.05) was found between the TP content and the DPPH antioxidant assay. More samples are needed to confirm this correlation.

### 2.2. Phenolic Compounds Identification in Persicaria maculosa Extracts by HPLC-ESI-MS/MS

The phenolic compounds were separated by HPLC and subsequently identified by ESI-MS/MS (the negative mode was used due to the acidic nature of phenols, and UV absorbance at 280 nm for recording of the chromatogram, Figure 1). The compounds were tentatively identified based on the UV spectra, mass fragmentation patterns, and comparison with the literature. A total of 27 compounds were tentatively identified (Table 3), including flavonol glycosides, methylated flavonols, and quinic acid (Figure 1 and Table 3). Among the chemical profile, quercetin glycosides are commonly reported in Polygonaceae [4], but myricetin glycosides and acylated flavonoids are unusual. A similar composition of phenolic compounds was reported in extracts from the aerial parts of *Polygonum equisetiforme* [30].

Peak 1 with a pseudomolecular ion at *m*/*z* 191 was identified as quinic acid. Peak 2 showed a pseudo-molecular ion at *m*/*z* 639, a quercetin-3-*O*-glucoside MS^2^ ion at *m*/*z* 463, and a quercetin MS^3^ ion at *m*/*z* 301, and thus was identified as quercetin-3-*O*-hexosyl-glucuronide, possibly quercetin-3-*O*-glucosyl-6”-*O*-glucuronide. In the same way, peak 3 was identified as quercetin 3,4’-di-*O*-glucoside (quercetin 3-*O*-neohesperidoside), [23] and peak 4 as myricetin 3´-glucose [31]. Peak 5, with a deprotonated molecular ion at *m*/*z* 609, was identified as kaempferol-*O*-di-glucose, producing intermediate MS^n^ ions at 429 and 285 (kaempferol). The isomer compounds at peaks 6 and 7 detected with a deprotonated molecular ion 609 were identified as quercetin-3-*O*-rutinose and quercetin-*O*-rhamnosyl-hexose, the last compound producing an MS^2^ ion at *m*/*z* 447 (quercetin rhamnose moiety) and a MS^3^ quercetin ion at *m*/*z* 301. Peak 8 with a pseudo-molecular ion at *m*/*z* 579 produced a daughter ion at *m*/*z* 285, and thus was identified as kaempferol-*O*-pentosyl glucose. Peak 9 was identified as quercetin-*O*-glucose. Peak 10, with a parent ion at *m*/*z* 593 and daughter kaempferol ion at *m*/*z* 285, was identified as kaempferol-*O*-rhamnosyl-glucose. Peak 11, which showed a parent ion at *m*/*z* 433, losing a 132-Dalton fragment, indicative of a pentose, was identified as quercetin-*O*-pentose, possibly quercetin-*O*-ribose or quercetin-*O*-arabinose [32]. Peak 12 had a deprotonated ion at *m*/*z* 477 and a diagnostic daughter ion at *m*/*z* 301, characteristic features of quercetin-*O*-glucuronide. In the same manner, peak 15 showed the characteristic fragmentation pattern of kaempferol-*O*-glucuronide [33]. The isomer compounds at peaks 13 and 14 showed deprotonated ions at *m*/*z* 447 but different daughter ions at *m*/*z* 285 and 301, characteristic of kaempferol-*O*-glucoside [34] and quercetin-*O*-rhamnose [35], respectively. Peak 16, with a deprotonated ion at *m*/*z* 563, showed a daughter ion at *m*/*z* 285 (loss of 278 Dalton, characteristic of rhamnosyl pentoside) and thus was identified as kaempferol-3-*O*-rhamnosylpentose. In the same manner, peak 20 was identified as kaempferol-3-*O*-galloyl-hexose and peak 21 as isorhamnetin-3-*O*-galloyl-hexose—possibly, these two peaks are kaempferol-3-*O*-(6´´*O*-galloyl)-glucose and isorhamnetin-3-*O*-(6´´*O*-galloyl)-glucose, respectively (Figure 2) [36]. Peaks 22 and 23, both with [M − H]^−^ ions at *m*/*z* 609 and daughter ions at *m*/*z* 323 and 285, were identified as two isomers of kaempferol-*O*-di-hexosides—possibly kaempferol-3-O-rutinose and kaempferol-3-*O*-sophorose, respectively [37]. Peak 24 was an isorhamnetin-*O*-sophorose, possibly isorhamnetin-3,4´di-*O*-diglucoside or isorhamnetin-*O*-di-hexoside [38]. Finally, peaks 25, 26, and 27 were identified as quercetin, isorhamnetin, and kaempferol, respectively. It is worth noting that the aglycones quercetin and isorhamnetin were not detected in either of the methanolic extracts, which probably indicates lower amounts in the plants. In addition, the aglycones have been reported as better antioxidants than their corresponding glycosides [39].

### 2.3. Antifeedant Activity

The antifeedant activities of the *Persicaria maculosa* extracts were evaluated for the first time against chewing and sucking insects. In order to have a broad spectrum of the anti-insect activities, four species were chosen as models. *Epilachna paenulata* (Coleoptera: Coccinellidae) was investigated because is a folivorous South American pest and feeds mainly on Cucurbitaceae species [40]. Meanwhile, *Pseudaletia adultera* (Lepidoptera: Noctuidae) was investigated because it is an important pest of the cultivated grasses also in South America [41]. In the same way, *Macrosiphum euphorbiae* (Hemiptera: Aphididae) is an important worldwide pest of potato crops [42] and *Diaphorina citri* (Hemiptera: Liviidae) is the principal pest of citrus and is considered an important vector of plant pathogenic proteobacteria [43].

#### 2.3.1. Antifeedant Activity against Chewing Insects (*E. paenulata* and *P. adultera*)

All the extracts exhibited a feeding deterrence at the end of the assays for both chewing insects, *E. paenulata* and *P. adultera* (Wilcoxon Signed-Rank tests, *p* < 0.05 at either 1-tail or 2-tail tests, as shown in Table 4). Among the extracts tested, the highest activity was for the Valparaíso extracts, especially when tested against *E. paenulata*. The Curicó extracts were the least effective deterrent agents against *P. adultera.* The activity as a function of time continually showed the same tendency, as for both insects the consumption on the control leaves was always higher than on the treated leaves (see Appendix A for *E. paenulata* consumption and Appendix A for *P. adultera* consumption in Appendix A). Similar studies have been conducted using extracts and isolated compounds from *Melia azedarach* fruits against *E. paenulata* [44].

#### 2.3.2. Antifeedant Activity against Sucking Insects (*Diaphorina citri*)

Although there is a tendency of all the extracts in decreasing feeding by *D. citri*, the antifeedant activity was only significant for the MeOH extracts (Figure 3, ANOVA, Tukey *p* < 0.05) compared to their respective controls. At the same time, comparing the extracts from the two locations (in the case of EtOH and MeOH extracts), the extracts from Curicó exhibited a higher decreasing effect on th eexcreted droplets (Figure 3, ANOVA, Tukey *p* < 0.05) (on a side note, we were able to verify that the addition of dimethyl sulfoxide (DMSO) did not have any effect on the *D. citri* excretion (*p* > 0.05)). These results may suggest a different composition of the plant material in both locations.

#### 2.3.3. Anti-settling Activity against Sucking Insects (*Macrosiphum*
*euphorbiae*)

*M. euphorbiae* preferred to settle on the control leaves over the extract-treated leaves in all cases, except for the EtOH and MeOH extracts from Curicó (Table 5, Wilcoxon signed-rank tests with *p* < 0.05 in all cases where significant differences were found), suggesting a general tendency where the aphids prefer control discs regardless of the extract offered vis-a-vis to the control.

All the extracts exhibited anti-settling activities; however, the DCM extract from Curicó clearly exhibited a higher anti-settling activity than the more polar extracts from the same location (Table 5, Kruskal–Wallis test, *p* = 0.001). These results agree with previous reports on the antifeedant activity from non-polar waxy extracts [45,46,47].

Although, in the case of the Valparaiso extracts, the PI seems also to increase from more polar extracts to the DCM extract, no significant differences were detected among the Valparaiso extracts (Kruskal–Wallis test, *p* = 0.1). Lastly, the DCM extracts from the two locations exhibited a similar anti-settling activity against *M. euphorbiae* (Kruskal–Wallis test, *p* = 0.6).

For chewing insects, a moderate correlation (*r* = 0.6338, *p* > 0.05) was found between the TP content and the IFP value for *P. adultera*. In the same manner, and inverse correlation (*r* = −0.8181, *p* > 0.05) was for the TP and IFP values of *E. paniculata*. Finally, in the case of *M. euphorbiae* an inverse correlation was obtained (*r* = −0.8917, *p* > 0.05). A comparative analysis of the antifeedant properties of the plants is explained below. Additionally, the antifeedant activities and the relation with phenolic compounds have been described. Quinic acid, ferulic acid, and related phenolic compounds contained in the methanolic extract from *Hyoscyamus muticus* L. exhibited antifeedant potential against the larvae *Spodoptera littorali* [48]. The well-known flavonoid, quercetin, isolated from the bark of *Bobgunnia madagascariensis* showed antifeedant activity against the beetle *Tribolium casteneum* [49]. Moreover, a high antifeedant activity has been reported against the termite *Coptotermes formosanus* Shiraki, mediated by flavonoids with hydroxyl groups at C-5 and C-7 in A-rings, such as kaempferol, quercetin, and myricetin [50]. Quercetin, isorhamnetin, and kaempferol 3-*O*-rutinoside derivatives and the flavonoid 5-hydroxy-3,7-dimethoxyflavone-4′-*O*-β-glucopyranoside isolated from the methanolic extract of *Calotropis procera* displayed a remarkable toxicity against the Coleoptera pests *Sitophilus oryzae* and *Rhyzopertha dominica* [51]. Previous studies have reported that the methoxyflavones 5-hydroxy-3,6,7,8,4‘-pentamethoxyflavone, 5-hydroxy-3,6,7,8-tetramethoxyflavone and 5,6-dihydroxy-3,7-dimethoxyflavone from cudweed *Gnaphaliumaffine* D. Don displayed a strong antifeedant activity against the moth larvae *Spodoptera litura* [52]. Interestingly, *Persicaria maculosa* extracts have the presence of different bioactive phenolic compounds, which could prove in part their antifeedant effects.

In summary, the results have shown that *P. maculosa* extracts are active and deterrent against insects such as *P. adultera*, *E. paenulata*, *D. citri,* and *M. euphorbiae*. The results observed in feeding deterrence trials could be related by the presence of bioactive quercetin, rutin, and kaempferol [53], besides other phenolic constituents. The glycosyl derivatives of those flavonoids are known for their antifeedant and insecticide activity [54,55] and were found in the *P. maculosa* polar extracts. Among the different flavonoids identified by HPLC-ESI-MS/MS in our study (Table 3), some of them contain methyl-ether groups in the A and C-ring (such as isorhamnetin), which could contribute to the antifeedant activity [56].

## 3. Materials and Methods

### 3.1. Plant Material

The aerial parts of *Persicaria maculosa* were collected by hand in February 2015 in Tranque La Luz, Valparaíso (Chile) (33° 7’ 7.23” S, 71° 34’ 47.23” W), and Quechereguas, Curicó (Chile) (35° 6 ‘55.12 “S, 71° 16’ 53.15” W). The plant was identified by Dr. Atala from the Pontificia Universidad Católica de Valparaíso (PUCV). The voucher specimens (No. 183592 and 183591) were deposited at the Herbarium of the Universidad de Concepción.

### 3.2. Extracts Preparations

The samples were washed and dried at 40 °C for 72 h in a stove. The dried material was ground and sieved, obtaining a particle size of 38 mesh. About 15–30 g of each sample was macerated with 80% MeOH, 80% EtOH, and dichloromethane (DCM) for 72 h at room temperature. The extracts were filtered, evaporated under reduced pressure (Büchi B-480, Büchi Labortechnik AG, Flawil, Switzerland) and stored in the dark at 4 °C until analysis.

### 3.3. Total Phenolic (TP) and Total Flavonoid (TF) Content

The TP content was determined by the Folin–Ciocalteu method with modifications [57]. Stock solutions of 5 mg/mL of all extracts under study were prepared in MeOH or MeOH: H_2_O 1:1 (*v*/*v*). An amount of 1 mL of this stock was mixed with 0.2 mL of Folin–Ciocalteu reagent and the volume was completed to 20 mL with distillated water. Then, 1 mL of Na_2_CO_3_ (20% *w*/*v*) was added and thoroughly mixed. Distillated water was added up to 25 mL and the mixture was incubated at room temperature in the dark for 1 h. The absorbance was measured at 725 nm in a Genesys 10UV spectrophotometer (ThermoSpectronic, Waltham, MA, USA).

The TF content was determined with the AlCl_3_ method [58]. In this assay, 0.25 mL of the same stock solution mentioned above was mixed with 75 μL of NaNO_2_ (5% *w*/*v*), thoroughly mixed in a vortex, and let to rest for 5 min. Then, 100 μL of AlCl_3_ (10% *w*/*v*) was added, thoroughly mixed in a vortex, and let to rest for additional 5 min. At the end of the incubation, 500 μL of NaOH (4% *w*/*v*) was added and immediately completed to 5 mL with distilled water. The mixture was left to rest at room temperature in the dark for 30 min and the absorbance was measured at 510 nm.

### 3.4. Antioxidant Activity

The antioxidant activity of the extracts (MeOH, EtOH and DCM) was evaluated by means of the following assays: the discoloration of the DPPH radical, the scavenging of the superoxide anion, the ferric-reducing antioxidant power (FRAP), and the cupric-reducing antioxidant power (CUPRAC). Stock solutions of 300 μg/mL of the extracts under study were used in the individual experiments with serial dilutions and in triplicate.

The free radical scavenging activity of the extracts was determined by the scavenging of the free radical DPPH and the superoxide anion, as previously described [57,59]. MeOH was used as the negative control and catechin was used as the reference compound. For the DPPH assay, the samples were dissolved in MeOH to a final concentration of 100 μg/mL and subsequently diluted in 96-well plates. Then, 200 μL of a 20 mg/L DPPH solution, freshly prepared in MeOH, was added. After 5 min, the absorbance was measured at 517 nm in a microplate reader (Biotek ELx800, Winooski, VT, USA). The superoxide anion scavenging assay was carried out as previously described [59], using Xanthine oxidase (X1875, EC 1.17.3.2) and hypoxanthine as the enzyme and substrate, respectively. Chromophore nitroblue tetrazolium salt (NBT) was used and the absorbances were measured at 560 nm.

The radical scavenging capacity was calculated using the following equation:Scavenging effects (%) = [A0 – A1/A0 × 100],
where A0 and A1 correspond to the absorbance of the radical in the absence and presence of antioxidant, respectively. The concentration of the extract that scavenges the free radical by 50% (SC_50_) is expressed as µg/mL and was calculated using the OriginPro 8.0 software (OriginLab Corporation, Northampton, MA, USA).

The FRAP and CUPRAC assays were performed as previously described [60]. Quantification was performed using a standard curve of the antioxidant Trolox. The results are expressed as mmol of Trolox equivalents (TE) per g of extract.

For all the experiments, stock solutions of 300 μg/mL of the sample were prepared. From this stock solution, serial dilutions (300, 200, 100, 50, 25, 12.5, and 6.3 μg/mL) were evaluated in triplicate in independent experiments. The results presented correspond to the mean values obtained ± the standard deviation of the triplicate (SD).

### 3.5. HPLC-ESI-MS/MS Analysis

An HPLC-ESI-MS/MS system consisting of an HP1100 chromatograph (Agilent Inc. Technologies, Santa Clara, CA, USA) coupled with a mass spectrometer Esquire 4000 Ion Trap LC/MS (n) system (Bruker Daltonik GmbH, Bremen, Germany) were employed for analysis. For the HPLC chromatograph control, the ChemStation LC 3D Rev. A.10.02 (Agilent Technologies Inc., Santa Clara, CA, USA) software was used; for the for the spectrometer control, the esquire Control 5.2 (Bruker Daltonik GmbH, Bremen, Germany) software was used. The ionization process (nebulization) by electrospray was conducted at 3000 V, assisted by nitrogen as the nebulizer gas (50 psi and 10 L/min flow) and assisted by nitrogen as the drying gas at 365 °C. All the experiments were carried out in negative mode. A C18 column of 250 × 4.6 mm, 5 μm, and 100Å was used for the HPLC separation (Luna, Phenomenex Inc., Torrance, CA, USA). The analysis conditions using a linear gradient contained 0.1% formic acid (A) and water (B): 0–5 min, 95%–5% B; 5–25 min, 60%–40% B; 25–30 min, 40%–60% B; 30–40 min, 20%–80% B; 40–50 min, 0%–100%; 50–55, 95%–5%. The flow rate was 1.0 mL/min, using a wavelength of 270 nm.

### 3.6. Insects

*Pseudaletia adultera* (Schaus) (Lepidoptera: Noctuidae, Hadenini) is a polyphagous insect which prefers grass. *P. adultera* came from a laboratory colony that was established from field-collected larvae from southern Uruguay of INIA (La Estanzuela, 34°20´23” S, 57°41´39” O). At the laboratory, the colony was fed with an artificial diet based on lima beans (100 g), dry yeast (15 g), wheat germ (50 g), maize bran (50 g), sorbic acid (1 g), ascorbic acid (3 g), and agar (18 g) in 400 mL of water. The controlled conditions were at 24 ± 1 °C, with a relative humidity of 65% and a photoperiod of (14 L: 10 d) in a growth house.

*Epilachna paenulata* Germar (Coleoptera: Coccinellidae) is an oligophagous insect specialized in cucurbitaceae. The adults were kept in the laboratory on pumpkin (*Cucurbita moschata*, Cucurbitaceae) using controlled temperature conditions (20 ± 2 °C) and photoperiod (14 L: 10 d).

Initial settlement of the laboratory colony was done from insects collected on organic-produced squash in Canelones, Uruguay (34°63´43” S, 56°04´45” O)

*Diaphorina citri* (Hemiptera: Liviidae) is a citrus pest. Adult individuals of *D. citri* were collected from a laboratory colony maintained in a greenhouse at the Estación Experimental INIA Salto Grande, Instituto Nacional de Investigación Agropecuaria, Salto, Uruguay. The culture was established from a field population collected in Salto, Uruguay (31°23′18″ S, 57°57′38″ W). The insects were reared on sweet orange (*Citrus sinensis*, Rutaceae) and Cravo lemon (*Citrus limon*, Rutacaea) potted seedlings in mesh cages (46 × 46 × 56 cm) at 14–28 °C, 73% HR, and a natural photoperiod.

*Macrosiphum euphorbiae* (Hemiptera: Aphididae) is a potato specialist. The aphid was raised on *Solanum commersonii* (Solanaceae) in a growth house at 19 ± 1 °C and a relative humidity of 60% ± 10%, with a photoperiod of (14 L: 10 d). The laboratory colony was established from insects collected from experimental potato crops of INIA (La Estanzuela, 34°20´23” S, 57°41´39” O).

### 3.7. Antifeedant Bioassay (Choice Experiment)

The antifeedant activity against *P. adultera* and *E. paenulata* was studied using two disks of the leaf (1 cm^2^), one for the control (C) and the other for the treatment (T). The leaf disks were cut from healthy plants of *Hordeum vulgare* and *Curcubita moschata*, for *P. adultera* and *E. paenulata*, respectively. Plant disks, lying in a 2% agar plate, were placed in an equidistant form in the petri dish (9 cm × 1 cm). The treatment disks (T) were covered with a 10 μL of the extract; the initial solution was (2 mg/100 μL) in acetone, while the control disks (C) were treated with 10 μL of solvent. The larvae in the third stage were individually assayed—*E. paenulata* (10 replicates) and *P. adultera* (15 replicates). To measure the antifeedant activity, a visual punctuation of the consumed area was assigned to each disk (0, 12.5, 25, 37.5, 50, 62.5, 75, 87.5, or 100%). For the activity, the data were analyzed by Wilcoxon signed-rank tests using the online resource http://vassarstats.net/, comparing the consumption on the control and treatment leaves. The data are presented as the index of feeding preference (IFP) calculated as IFP = (C − T)/(C + T), where C and T are the amounts consumed on the control and treatment leaves, respectively (the activity is feeding deterrent when the IFP > 0). Besides this, the data as function of time included in the Appendix A were analyzed by ANOVA with repeated-measures and the differences between the options were established by the Tukey-HSD proof using the Statgraphics Centurion XV (Statpoint, Inc., Warrenton, VA, USA) software. All the results are expressed as mean and standard error of the mean (SEM) values [61].

### 3.8. Aphid Preference Bioassays (Choice Experiment)

The bioassays were conducted with the aphid *M. euphorbiae*. Leaf disks (1.2 cm diameter) were cut from healthy *S. commersonii* plants and set flat over a layer of 2% agar, equidistant from the center and the margins of a Petri dish (6.1 cm diameter). In the experiment, the insects were offered a choice between leaf disks of *S. commersonii* with solvent as a control and one to evaluate the effect of the extracts. The leaf disks of *S. commersonii* were treated with the topical addition of extracts from *P. persicaria*. The EtOH extract, MeOH extract, and DCM extract (2 mg) were added to 100 μL acetone to obtain a dosage of 10 μL. The aphid preference was evaluated as the percentage of aphids settled on each leaf disk after 24 h. Twenty *M. euphorbiae* (adult and late instar nymphs) were placed in the center of the Petri dish using a soft paint brush and the dish was left upside down for 24 h. The replicates were considered valid if more than 50% of the aphids were alive after 24 h, and if at least half of these had settled on a leaf disk. The insect preference in the choice bioassays was analyzed by the Wilcoxon signed-rank test for paired samples using the online resource http://vassarstats.net/. Differences were considered significant at *p* < 0.05.

Data are presented as the preference index (PI) calculated as PI = (%C − %T)/(%C + %T), where %C is the % of aphids settled on the control and %T is the percent on the treated leaf (the activity is settling deterrent when PI > 0). All the data are expressed as mean ± SEM throughout [62].

### 3.9. Antifeedant Activity Evaluation on D. citri Adults (Choice Experiment)

To determine whether the plant extracts’ presence on citrus leaves affected the feeding of *D. citri* adults, a leaf contact assay was used [63]. Disks (5.5 cm diameter) were cut from freshly excised Eureka lemon (*Citrus limon*) leaves and placed on 3% agar beds in 5.5 mm diameter plastic disposable petri dishes. Solutions (13.3 mg/mL) were prepared from the dry residue of different plant extracts. DMSO (0.67 *v*/*v*) was added to MeOH, EtOH, and DCM extracts to enhance the dilution in water. Each disc leaf was treated with 150 µL of the solutions (total amount applied: 2 mg in 20 cm^2^). A 0.67 *v*/*v* aqueous solution of DMSO was used as a control, which was previously tested not to have a significant effect on *D. citri* feeding. The leaf discs were aired until dried. Subsequently, 6 *D. citri* adults were released into each dish and dishes were closed. The insects were collected in a glass tube (4 mL) that were cooled (−4 °C, 2 min) before the release. The inside part of the cap of each plate was completely covered with a 5.5 cm filter paper (Whatman type, Macherey Nagel GMBH, Düsseldorf, Germany). Petri dishes placed upside down to collect excreted honeydew droplets on the filter papers, and were maintained at 22 ± 2 °C and 45 ± 5% RH under a 14:10 h light: dark photo cycle. At 24 h after the insect release, the filter papers were collected and soaked for approximately 3 min an acetone solution of ninhydrin (1% *w*/*v*, Sigma-Aldrich). The ninhydrin turned honeydew droplets on filter paper discs into dark purple spots. The number of these spots on each filter paper was counted. Two sets of experiments were carried out, one for each location—the Valparaíso and Curicó extracts. On each set, the control measurements were included. For each set, one dish was considered a replicate and 10 replicates were set for each treatment. The 10 replicates were carried out within 3 different dates, which were considered as blocks in the statistical analysis. Droplets count was log-transformed to adjust the data to a normal distribution and afterwards subjected to an analysis of variance (ANOVA). The treatment means were compared using Tukey’s test at an α = 0.05. The free Infostat software package (infostat, Cordoba, Argentina) was used.

## 4. Conclusions

The ethanolic and methanolic extracts obtained from two samples of *P. maculosa* collected in Valparaíso and Curicó, were analyzed by HPLC-ESI-MS/MS and showed the presence of quinic acid plus 26 flavone compounds, mainly *O*-glycosylated flavones. The antioxidant activity showed a difference among ethanolic and methanolic extracts, with the species collected in Curicó shown to be the most active in the FRAP, DPPH, and O_2_^−^ scavenging assays, which might be related to the moderate deterrent activity against the insects studied. However, more studies are needed since no clear correlations were detected among the chemistry and the bioactivity results. The antifeedant activity against the chewing insects (the coleoptera *E. paenulata* and lepidoptera *P. adultera*) was significant for all samples, with the activity highest being against the specialist *E. paenulata*. In the case of sucking insects, only the MeOH extracts from both locations were active feeding deterrents against *D. citri*; in contrast, in the case of *M. euphorbiae* the less non-polar extract (DCM) was the most active. These results illustrate the importance of using more than one insect model when trying to characterize the deterrent effects from plant extracts. The bioassay-guided fractionation and further isolation of the main compounds is needed to determine the bioactive compounds responsible for these activities.

## Figures and Tables

**Figure 1 molecules-25-03054-f001:**
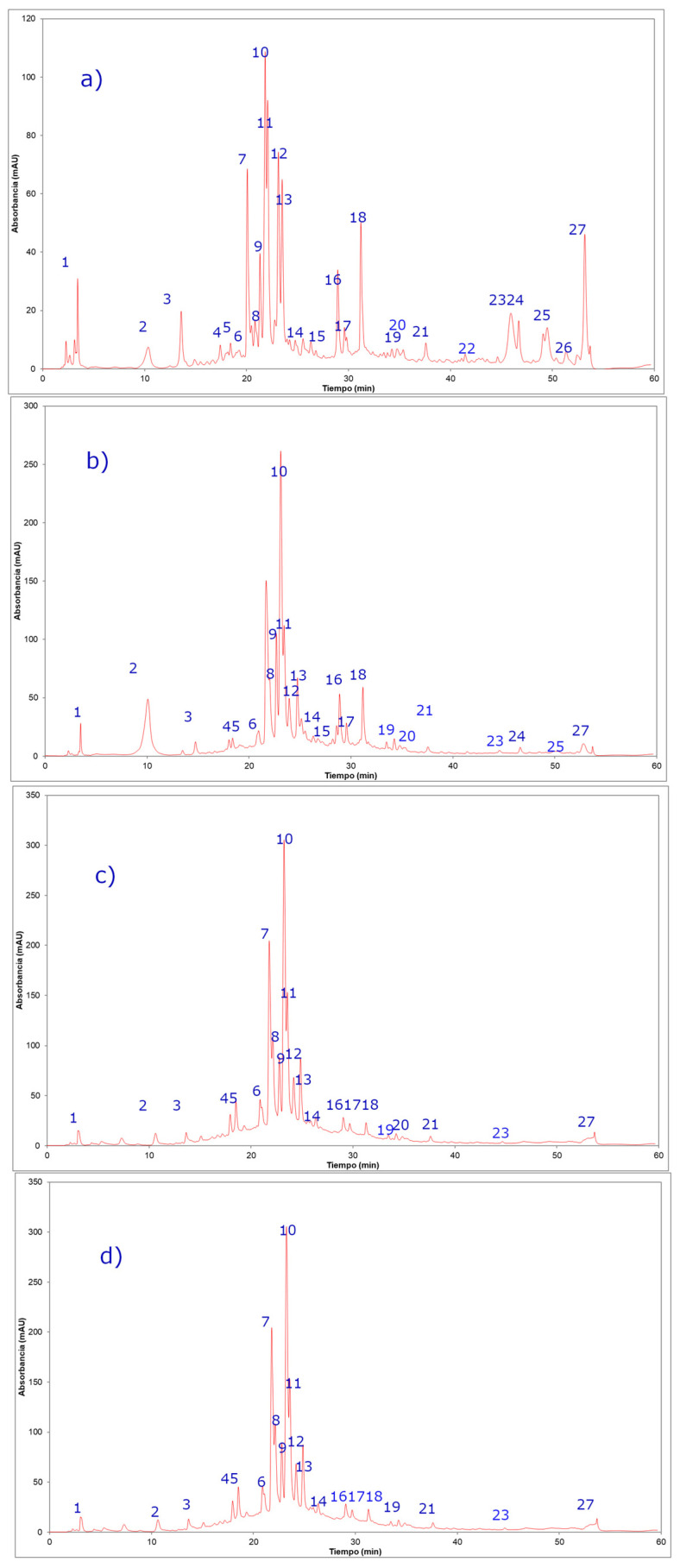
HPLC photodiode assay detector (PDA) chromatograms of *Persicaria maculosa* extracts (spectra at 280 nm). (**a**) Curicó ethanolic extract, (**b**) Valparaíso ethanolic extract, (**c**) Curicó methanolic extract, (**d**) Valparaíso methanolic extract.

**Figure 2 molecules-25-03054-f002:**
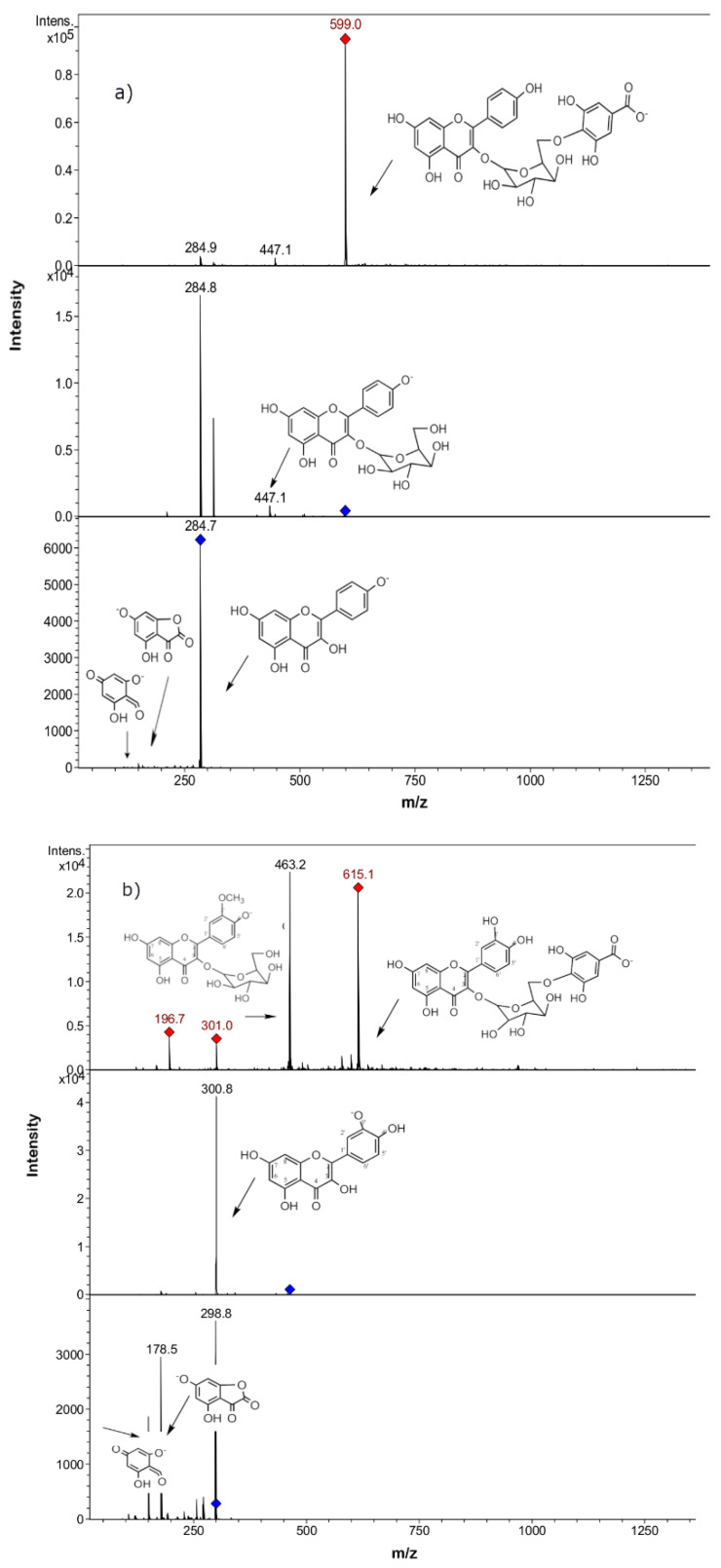
Full MS (upper panel) and MS1 (medium panel) and MS2 (lower panel) spectra of compound **20**, tentatively identified as Kaempferol-*O*-galloyl-hexose, (**a**) and compound **21**, tentatively identified as Isorhamnetin-*O*-galloyl-hexose (**b**).

**Figure 3 molecules-25-03054-f003:**
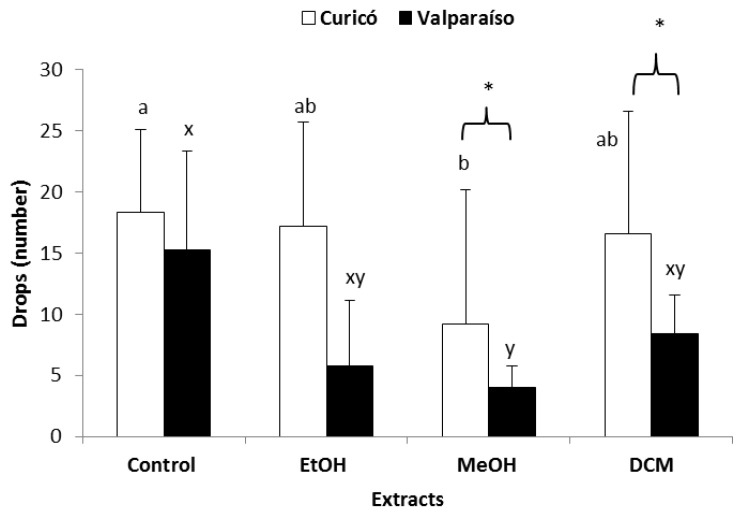
Antifeedant activity on *Diaphorina citri* adults. Results are shown as means ± SD of honeydew droplets. Different letters indicate significant differences (ANOVA, Tukey *p* < 0.05) among the Curicó extracts (open bars; a, b) or among the Valparaiso extracts (solid bars; x, y). Asters indicate significant differences between the same kind of extracts from different locations (* ANOVA, Tukey *p* < 0.05).

**Table 1 molecules-25-03054-t001:** Total phenolics and total flavonoids of *Persicaria maculosa* extracts (*N* = 3/extract) ^*^.

*P. maculosa* Extracts	Total Phenolics (g GAE/100 g Extract)	Total Flavonoids (g CE/100 g Extract)
EtOH extract from Curicó	16.6 ± 0.2 ^a^	8.4± 0.1 ^a^
EtOH extract from Valparaíso	13.7 ± 0.5 ^b^	6.6 ± 0.2 ^b^
MeOH extract from Curicó	14.3 ± 0.9 ^b^	8.2 ± 0.2 ^a^
MeOH extract from Valparaíso	9.6 ± 0.2 ^c^	8.6 ± 0.2 ^a^

* Results are shown as means ± SD. Different superscript letters (a–c) in the same column show significant differences among the samples, according to ANOVA with ad hoc Tukey’s test (*p* < 0.05). GAE = Gallic Acid Equivalents, CE = Catechin Equivalents.

**Table 2 molecules-25-03054-t002:** Antioxidant activity of *Persicaria maculosa* extracts (*N* = 3/extract) ^*^.

*P. persicaria* Extracts	DPPH Scavenging SC_50_ (µg/mL)	O_2_^−^ Scavenging SC_50_ (µg/mL)	FRAP (mmol TE/g Extract)	CUPRAC (mmol TE/g Extract)
EtOH extract from Curicó	12.5 ± 0.2 ^a^	22.1 ± 0.6 ^a^	1.6 ± 0.1 ^a^	0.8 ± 0.1 ^a^
EtOH extract from Valparaíso	14.2 ± 0.2 ^a^^,b^	34.1 ± 3.2 ^b^	0.8 ± 0.1 ^b^	0.7 ± 0.1 ^a^
MeOH extract from Curicó	15.5 ± 1.1 ^b,c^	43.1 ± 1.6 ^c^	0.8 ± 0.1 ^b^	0.7 ± 0.1 ^a^
MeOH extract from Valparaíso	18.0 ± 1.6 ^c^	36.3 ± 1.1 ^b^	1.0 ± 0.1 ^b^	1.1 ± 0.1 ^b^
Catechin ^#^	11.4 ± 1.6	8.7 ± 0.1	5.4 ± 0.1	13.4 ± 0.3

* Results are shown as means ± SD (*N* = 3). # Catechin was used as the reference compound. Different superscript letters (a–c) in the same column show significant differences among the samples, according to Tukey’s test (*p* < 0.05). In the reducing power assays, the results were expressed as mmol of Trolox equivalents (TE) per g of extract.

**Table 3 molecules-25-03054-t003:** Tentative HPLC electrospray mass spectrometry (HPLC-ESI-MS/MS) identification of compounds in *Persicaria maculosa* extracts.

Peak	UV Max	Rt (min)	[M − H]^−^ Ion	Fragment Ions *(m*/*z)*	Tentative Identification	*Persicaria maculosa* Extracts *			
						C.E	V.E	C.M	V.M
1	205	3.3	191	170, 152, 126, 110, 84	Quinic acid ^b^	+	+	+	+
2	254–354	16.9	639	463(quercetin-3-*O*- hexose), 301, 179, 151	Quercetin-3-*O*-hexosyl-glucuronide ^b^	+	+	+	+
3	254–354	19.6	625	463(quercetin-3-*O*- hexose moiety), 301, 179, 151	Quercetin 3,4´-di-*O*-glucoside, (quercetin sophoroside) ^a^	+	+	+	+
4	254–354	20.3	479	317 (myricetyn), 179, 151	Myricetin 3´-glucose ^b^	+	+	+	+
5	264–364	20.4	609	590, 429, 285	Kaempferol-*O*-di-glucose ^a,b^	+	+	+	+
6	254–354	21.1	609	301(quercetin), 179, 151	Quercetin-3-*O*-rutinose (rutin) ^a^	+	+	+	+
7	254–354	21.1	609	447 (quercetin rhamnose), 301	Quercetin-*O*-rhamnosyl-hexose (Quercetin 3-*O*-neohesperidose) ^b^	+	+	+	+
8	264–364	21.4	579	285 (kaempferol), 179, 151	Kaempferol-*O*-pentosyl glucose ^b^	+	+	+	+
9	254–354	21.7	463	301(quercetin), 179, 151	Quercetin-*O*-glucose ^a^	+	+	+	+
10	264–364	22.3	593	285(kaempferol), 179,151	Kaempferol-*O*-Rhamnosyl-glucose ^b^	+	+	+	+
11	254–354	23.1	433	301(quercetin), 179, 151	Quercetin-*O*-pentose ^a,b^	+	+	+	+
12	254–354	23.3	477	301(quercetin), 179, 151	Quercetin-*O*-glucuronide ^a^	+	+	+	+
13	264–364	23.3	447	285 (kaempferol), 179, 151	Kaempferol-*O*-glucoside ^a,b^	+	+	+	+
14	254–354	23.5	447	301(quercetin), 271, 179, 151	Quercetin-*O*-rhamnose ^b^	+	+	+	+
15	254–354	24.2	461	285(kaempferol), 179, 151	Kaempferol-*O*-glucuronide ^b^	+	+	+	+
16	264–364	28.9	563	413, 285, 179, 151	Kaempferol-*O*-rhamnosyl pentose ^b^	+	+	+	+
17	264–364	30.1	417	285(kaempferol), 179, 151	Kaempferol-*O*-pentose ^a,b^	+	+	+	+
18	264–364	31.5	431	285(kaempferol), 179, 151	Kaempferol-*O*-rhamnose ^a,b^	+	+	+	+
19	264–364	32.3	431	285(kaempferol), 179, 151	Kaempferol-*O*-rhamnose ^a,b^	+	+	+	+
20	264–364	34.8	599	447 (kaempferol-3-*O*- hexose), 285	Kaempferol-*O*-galloyl-hexose ^a,b^	+	+	+	+
21	254–354	30.1	615	463, 301, 179, 151	Isorhamnetin-*O*-galloyl-hexose ^b^	+	+	+	+
22	264–364	41.6	609	323 (rutinose), 285, 179, 151	Kaempferol-*O*-rutinose ^a,b^	+	+	+	+
23	264-364	45.2	609	323 (sophorose), 285, 179, 151	Kaempferol-3-*O*-sophorose ^a,b^	ND ^c^	+	+	+
24	254–354	48.1	639	463(quercetin-3-*O*-hexose), 315(isorhamnetin-3-*O*-hexose), 301, 179, 151	Isorhamnetin-*O*-sophorose ^b^	+	+	ND	ND
25	254–354	50.2	301	194, 271, 179, 151	Quercetin ^a,b^	+	+	ND	ND
26	254–354	50.5	315	300, 179, 151	Isorhamnetin ^a,b^	+	+	ND	ND
27	264–364	53.4	285	179,151	Kaempferol ^a,b^	+	+	+	+

* CE = Curicó Ethanolic extract, VE = Valparaíso Ethanolic extract, CM = Curicó Methanolic extract, VM = Valparaíso Methanolic extract. Identification according to ^a^ literature from the family Polygonaceae, ^b^ fragmentation patterns. ^c^ ND, not detected.

**Table 4 molecules-25-03054-t004:** Antifeedant activity presented as the index of feeding preference (IFP) on *P. adultera* and *E. paenulata* larvae of the *P. maculosa* leaf extracts from Curicó and Valparaíso at the end of the assays. Results are shown as mean ± standard error (SE).

Insect Species	Extract	IFP (a)		Wilcoxon Signed-Rank Test Results	
		Curicó Extracts	Valparaíso Extracts	Curicó Extracts	Valparaíso Extracts
*Pseudaletia adultera* (b)	EtOH	0.6 ± 0.2 *	0.6 ± 0.2 *	W = −72; ns/r = 15; z = −2.03 P(1-tail) = 0.02; P(2-tail) = 0.04	W = −72; ns/r = 15; z = −2.03 P(1-tail) = 0.02; P(2-tail) = 0.04
	MeOH	0.5 ± 0.2 **	0.5 ± 0.2 **	W = −56; ns/r = 15, z = −1.58 P(1-tail) = 0.05; P(2-tail) = 0.1	W = −56; ns/r = 15; z = −1.58 P(1-tail) = 0.05; P(2-tail) = 0.1
	DCM	0.7 ± 0.1 *	0.7 ± 0.2 *	W = −66; ns/r = 11; z = −2.91 P(1-tail) = 0.002; P(2-tail) = 0.004	W = −88; ns/r = 15; z = −2.48 P(1-tail) = 0.007; P(2-tail) = 0.01
*Epilachna paenulata* (c)(d)	EtOH	0.6 ± 0.3 **	1.0 ± 0.0 *	W = −33; ns/r = 10; z = −1.66 P(1-tail) = 0.05; P(2-tail) = 0.1	W = −55; ns/r = 10; z = −2.78 P(1-tail) = 0.003; P(2-tail) = 0.005
	MeOH	0.8 ± 0.2 *	1.0 ± 0.0 *	W = −44; ns/r = 10; z = −2.22 P(1-tail) = 0.01; P(2-tail) = 0.03	W = −55; ns/r = 10; z = −2.78 P(1-tail) = 0.003; P(2-tail) = 0.005
	DCM	0.8 ± 0.2 *	1.0 ± 0.0 *	W = −44; ns/r = 10; z = −2.22 P(1-tail) = 0.01; P(2-tail) = 0.03	W = −55; ns/r = 10; z = −2.78 P(1-tail) = 0.003; P(2-tail) = 0.005

(a) IFP = (C − T)/(C + T), where C and T are the amounts consumed on the control and treatment leaves, respectively, for the chewing insects (the activity is feeding-deterrent when the IFP > 0). (b) *N* = 15/extract, final time = 120 min; (c) *N* = 10/extract, final time = 180 min (Curicó extracts).; (d) *N* = 10/extract, final time = 135 min (Valparaíso extracts). * Denotes significant difference between the C and T consumption (deterrent, *p* < 0.05, 2-tail test); ** denotes significant difference between the C and T consumption (deterrent, *p* < 0.05, 1-tail test); NS denotes no significant differences.

**Table 5 molecules-25-03054-t005:** Anti-settling activity ^(a)^ on *Macrosiphum euphorbiae* of the different extracts (*N* = 10 replicate/extract; 20 aphids/replicate).

	Valparaíso Extracts			Curicó Extracts		
Type of Extract	% Settling on Treated Leaf	% settling on control leaf	PI ^(a)^	% Settling on Treated Leaf	% Settling on Control Leaf	PI
EtOH	34 ± 4	66 ± 4	0.32 ± 0.08 *	44 ± 3	56 ± 3	0.12 ± 0.06 (NS)
MeOH	31 ± 3	69 ± 3	0.37 ± 0.06 *	41 ± 5	59 ± 5	0.18 ± 0.09 (NS)
DCM	23 ± 4	77 ± 4	0.55 ± 0.08 *	23 ± 5	79 ± 5	0.58 ± 0.09 *

^(a)^ Results are shown as mean ± SE. PI is the preference index, calculated as (%C − %T)/(%C + %T), where %C is the % of aphids settled on the control and %T on the treated leaf (the activity is settling deterrent when PI > 0). * Shows significant differences between settling on the control and treated leaves (Wilcoxon signed-rank test, *p* < 0.05 in all cases); NS stands for non-significant differences.

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
