# Peer review of "Phenolic Fingerprinting, Antioxidant, and Deterrent Potentials of Persicaria maculosa Extracts"

_molecules, 2020, doi:10.3390/molecules25133054_

Round 1

Reviewer 1 Report

Reviewer #1:

Phenolic Fingerprinting, Antioxidant, and Deterrent Potentials of Persicaria maculosa Extracts

This paper focuses on chemical profiling by HPLC and antioxidant activities of two populations of Persicaria maculosa extracts from two regions of Chile were reported.  The authors reported the presence of quinic acid plus 26 flavonoid compounds. Additionally, four insect species were chosen to study the potential deterrent activities based on their different feeding habits, diet breadth, and host plants. No correlations were detected among the chemistry and the bioactivity results. This is a relatively simple study, however, the article appears to be interesting to Molecules readers. The article is not well-structured, and the material and methods are not well-defined. Significant English editing is needed.

Authors need to define and standardize the abbreviations on the first mention in the abstract, keywords, main text, and tables/figures.

Please provide more information regarding how to express the results of antioxidant activity in (SC50). Please complete the method of antioxidant activity by DPPH. Please check again, which is correct between SC50 or IC50 for this method, because these are different meaning and usage.

Table 4 is not self-explanatory and deserves greater attention.

My comments and judgment are in the pdf file (highlighted in yellow).

Reviewer 2 Report

The work studies phenolic fingerprinting, antioxidant, and deterrent potentials of Persicaria maculosa extracts. The results showed that P. maculosa is a good source of flavonoids with some antioxidant capacities and have potential interest as botanical eco-friendly alternative with deterrent activity.

The paper is interesting and concerns a very current topic, however there are some points that need to be clarified or modified before publication.

In my opinion authors should perform quantitative HPLC analysis of inestigated compounds. What is more, the work abounds in grammatical, stylistic, spelling errors and typos.

Reviewer 3 Report

General comments:

Overall, this manuscript was clear. The aims, the structure and methodology were properly conducted. Before being considered for publication, authors must take into consideration the following comments:

  • Discuss the results in deep.
  • Clarify why two solvents were used for analyzing the antioxidant activity of extracts and three solvents in anti-feeding assays.
  • Clarify why the correlation was only made between TF - TP and the antioxidant assays if it will be more interesting to correlate the phenolic content of samples and the antifeedant activity & Anti-settling activity.

Abstract:

In the abstract is clear all the assays made for two populations of Persicaria maculosa but is unclear that extracts were made with 3 different solvents: methanol, ethanol and dichloromethane. Please, add this information for better understanding.

Results and discussion section:

Please, add the meaning of EtOH and MeOH the first time appear.

Tables 1 and 2, as well as text related with this table: What is about results related with dichloromethane?

Discussion of the results was not found.

Section 2.3: In my opinion, this section is more related with the introduction. Here the results obtained by authors are not presented.

Section 2.4: “treated leaves” instead of “treted leaves” at the end of the paragraph.

Material and methods section:

Section 3.2: Why the three solvents (methanol, ethanol and dichloromethane) were not used for all analysis? Please, explain it.

Section 3.3: This methodology is not “briefly” explained as stated by authors in the first line. Please, add information on this regard.

Was the DPPH assay only made with methanolic extracts?

Section 3.9: Please, add the meaning of DMSO.

Conclusion:

“Antifeedant” instead of “ñAntifeedant”.

Why the correlation was only made between TF - TP and the antioxidant assays? It will be more interesting to correlate the phenolic content of samples and the antifeedant activity & Anti-settling activity.

Round 2

Reviewer 3 Report

Dear authors,

The manuscript has been improved and you take into consideration almost all the suggestions. However, taking into consideration the that aim of the research was “The purpose of this study was to investigate the antifeedant potential of methanolic (MeOH), ethanolic (EtOH) and dichloromethane (DCM) extracts from the aerial parts of this plant collected in Valparaíso and Curicó provinces (Chile) and relate this activity to the antioxidant capacity and the presence of phenolic compounds in the extracts”, the correlation between phenolic compounds and the antifeedant activity, as well as anti-settling activity is one of the main topic of the research.

The phrase “For chewing insects, a moderate correlation (r= 0.6338) was found between the TP content and the IFP value for P. adultera. In the same manner, and inverse correlation (r= -0.8181) was for TP and IFP values of E. paniculata. Finally in the case of M. euphorbiae and inverse correlation was obtained (r= -0.8917)” was added but no discussion on this regard was found. Please, discuss in deep the meaning of this correlation.

Author Response

Reviewer: The manuscript has been improved and you take into consideration almost all the suggestions. However, taking into consideration the that aim of the research was “The purpose of this study was to investigate the antifeedant potential of methanolic (MeOH), ethanolic (EtOH) and dichloromethane (DCM) extracts from the aerial parts of this plant collected in Valparaíso and Curicó provinces (Chile) and relate this activity to the antioxidant capacity and the presence of phenolic compounds in the extracts”, the correlation between phenolic compounds and the antifeedant activity, as well as anti-settling activity is one of the main topic of the research.

The phrase “For chewing insects, a moderate correlation (r= 0.6338, p >0.05) was found between the TP content and the IFP value for P. adultera. In the same manner, and inverse correlation (r= -0.8181, p >0.05) was for TP and IFP values of E. paniculata. Finally, in the case of M. euphorbiae and inverse correlation was obtained (r= -0.8917, p >0.05)” was added but no discussion on this regard was found. Please, discuss in deep the meaning of this correlation.

Answer: We have carried out the discussion of the results, in the following line of text:

Line 202: The correlation analysis paragraph has been removed and moved to line 272

Line 272:  For chewing insects, a moderate correlation (r= 0.6338, p >0.05) was found between the TP content and the IFP value for P. adultera. In the same manner, and inverse correlation (r= -0.8181, p >0.05) was for TP and IFP values of E. paniculata. Finally, in the case of M. euphorbiae and inverse correlation was obtained (r= -0.8917, p >0.05). A comparative analysis of the antifeedant properties from plants is explained below. Additionally, the antifeedant activities and the relation with phenolic compounds has been described. Quinic acid, ferulic acid, and related phenolic compounds contained in the methanolic extract from Hyoscyamus muticus L. exhibit antifeedant potential against the larvae Spodoptera littorali [1]. Well-known flavonoid, quercetin, isolated from the bark of Bobgunnia madagascariensis showed antifeedant activity against the bettle Tribolium casteneum [2]. Has been reported high antifeedant activity against the termite Coptotermes formosanus Shiraki in flavonoids with hydroxyl groups at C-5 and C-7 in A-rings such as kaempferol, quercetin and myricetin [3]. Quercetin, isorhamnetin and kaempferol 3-O-rutinoside derivatives, and the flavonoid 5-hydroxy-3,7-dimethoxyflavone-4′-O-β-glucopyranoside isolated from the methanolic extract of Calotropis procera displayed remarkable toxicity against the Coleoptera pests Sitophilus oryzae and Rhyzopertha dominica [4]. Previous studies reported that the methoxyflavones 5-hydroxy-3,6,7,8,4‘-pentamethoxyflavone, 5-hydroxy-3,6,7,8-tetramethoxyflavone and 5,6-dihydroxy-3,7-dimethoxyflavone from cudweed Gnaphaliumaffine D. Don displayed strong antifeedant activity against the moth larvae Spodoptera litura [5]. Interestingly, Persicaria maculosa extracts have the presence of different phenolic compounds, that could prove in part its antifeedant effects.

  1. Eman, R.E.; Abdelaziz, E.; Emad, M.A.; Ahmed, M.H.A. Antioxidant, antimicrobial and antifeedant activity of phenolic compounds accumulated in Hyoscyamus muticus L. African J. Biotechnol. 2018, 17, 311–321.
  2. Adebote, D.; Amupitan, J.; Oyewale, A.; Agbaji, A. Antifeedant Activity of Quercetin Isolated from the Stem Bark of Bobgunnia madagascariensis (Desv.) J.H.Kirkbr & Wiersema. (Caesalpiniaceae). Aust. J. Basic Appl. Sci. 2010, 4, 3342–3346.
  3. Ohmura, W.; Doi, S.; Aoyama, M.; Ohara, S. Antifeedant activity of flavonoids and related compounds against the subterranean termite Coptotermes formosanus Shiraki. J. Wood Sci. 2000, 46, 149–153.
  4. Nenaah, G.E. Potential of using flavonoids, latex and extracts from Calotropis procera (Ait.) as grain protectants against two coleopteran pests of stored rice. Ind. Crops Prod. 2013, 45, 327–334.
  5. Morimoto, M.; Kumeda, S.; Komai, K. Insect antifeedant flavonoids from Gnaphalium affine D. Don. J. Agric. Food Chem. 2000, 48, 1888–91.
